# Characterization of Microscopic Multicellular Foci in Grossly Normal Renal Parenchyma of Von Hippel-Lindau Kidney

**DOI:** 10.3390/medicina58121725

**Published:** 2022-11-24

**Authors:** Nayef S. Al-Gharaibeh, Sharon B. Shively, Alexander O. Vortmeyer

**Affiliations:** 1Department of Pathology, Indiana University School of Medicine, 350 W 11th Street, Suite 4034, Indianapolis, IN 46202, USA; 2Department of Physiology and Biochemistry, Jordan University of Science and Technology, 22110 Al Ramtha, Jordan; 3Surgical Neurology Branch, National Institute of Neurological Disorders and Stroke, National Institutes of Health, Bethesda, MD 20894, USA; 4Department of Molecular Medicine, Institute for Biomedical Sciences, The George Washington University, Washington, DC 5246, USA

**Keywords:** 3dimensional reconstruction, von Hippel-Lindau disease, renal clear cells, kidney cancer, precursor structures

## Abstract

*Background and Objectives*: This study aims to describe the earliest renal lesions in patients with von Hippel-Lindau (VHL) disease, especially the multicellular microscopic pathologic events, to get information into the genesis of renal neoplasms in this condition. *Materials and Methods*: Multicellular events were identified, and 3dimensional reconstruction was performed in grossly normal kidney parenchyma from VHL disease patients by using H&E-stained slides previously prepared. *Results:* The lesions were measured and the volume of clusters was calculated. Immunohistochemistry was performed for downstream *HIF*-target protein carbonic anhydrase 9 (CAIX) as well as CD34 for assessment of angiogenesis. We divided lesions into four types according to lesion height/size. The number of lesions was markedly decreased from lesion 1 (smallest) to lesion 2, then from lesions 2 to 3, and again from lesion 3 to 4. Distribution was highly consistent in the four cases, and the same decrement pattern was seen in all blocks studied. The volumes of clusters were measured and divided into three categories according to their volume. The most frequent pathologic event in VHL kidneys was category 1 (smallest volume), then category 2, and then category 3. *Conclusion:* We demonstrate that tracking histologic and morphologic changes in 3 dimensions of multicellular microscopic pathologic events enabled us to confirm a protracted sequence of events from smaller to larger cellular amplification events in VHL kidney.

## 1. Introduction

Von Hippel-Lindau (VHL) disease was first described in 1926 [1]. VHL disease is an autosomal dominant inherited familial tumor syndrome caused by germline mutation of the VHL tumor suppressor gene [2]. The VHL gene is a tumor suppressor gene on the short arm of chromosome 3 (3p25–26) [3]. The VHL protein (pVHL) is expressed in fetal and adult human tissues. The lack of this protein induces profound intracellular metabolic changes that closely resemble changes observed in oxidative stress [4]. Knudson proposed in 1971 a two-hit hypothesis for VHL disease. The first hit, germline mutation in the VHL gene, leads to the inactivation of a tumor suppressor allele. The second hit refers to somatic mutation, which occurs after the first hit and leads to the inactivation of the second allele [5]. Loss of VHL function induces negative regulation of the two alpha-subunits of hypoxia-inducible factor (HIF). HIF-regulated genes include vascular endothelial growth factor (VEGF), carbonic anhydrase IX (CAIX), erythropoietin (EPO), platelet-derived growth factor (PDGF), and the glucose-transporter 1 (GLUT-1) [3,6]. Cells with a loss of VHL-mediated HIF degradation express higher HIF alpha proteins, VEGFA, and CAIX. Loss of pVHL results in a lower breakdown and higher concentration of HIF, which leads to more increased secretion of pro-angiogenic HIF target proteins [2,6]. HIF is significant for tumor persistence because it stimulates angiogenesis, which results from increased levels of VEGF, PDGF-beta, or both. VEGF and PDGF-beta are necessary for the proliferation of endothelial cells and pericytes, respectively [7].

Patients affected by VHL disease develop specific types of heavily vascularized tumors in a highly selective subset of organs. Multiple tumors occur frequently. These include retinal, cerebellar, brainstem and spinal cord hemangioblastoma, microcystic cystadenoma, epididymal cystadenoma, pheochromocytoma and paraganglioma, endolymphatic sac tumor, as well as renal clear cell carcinoma [8,9,10,11,12,13,14].

Renal clear cell carcinoma (RCCC) shares histopathologic features with hemangioblastic central nervous system tumors and epididymal tumors. It is characterized by the proliferation of clear cells with abundant reactive vascularization [6,15]. Mandriota et al. showed that HIF activation could be identified in both early and late lesions in kidneys of VHL patients. In addition, there was an increase in vascularization around tubules containing CAIX-immunoreactive cells [9]. Loss of heterozygosity (LOH) on chromosome 3p has been reported in macroscopic RCCC of VHL and sporadic tumors [16,17,18,19].

Although clinical, radiological, and pathological data suggested that the precursors of RCCC in patients with VHL disease could be a benign or atypical renal lesion [20,21,22], the definitive transformation has not been well documented in the literature thus far.

Lubensky et al. showed that 25 of 26 clear-cell renal lesions lost the wild-type allele, and that was the first molecular evidence that LOH of the normal VHL disease gene occurs in benign and atypical clear-cell cysts. This may represent an early event in the development and progression of RCCC in VHL [13].

Characterization of precursor structures has provided more precise insight into the anatomic and cytologic origin of neoplastic processes associated with VHL disease. Precursor structures are far more numerous than frank tumors. VHL inactivation after the “second hit” is necessary but insufficient to develop mass-forming tumors [11,23]. Comparative molecular analysis of tumors and precursor structures may be helpful to identify a “third hit” promoting tumorigenesis from precursor material [11].

This study characterizes the multicellular microscopic pathologic events using a novel 3-dimensional algorithm consisting of methodological sampling and histological tracking. We tracked lesions in 3 dimensions to confirm their relationship to each other. This algorithm allows us to observe that microscopic clear cell proliferations follow a predictable pattern covering a morphological spectrum of histologic lesions from an isolated clear cell to potentially tumorigenic clear cell proliferation.

## 2. Materials and Methods

### 2.1. Ethics Statement and Sample Collection

Tissue has been obtained from seven kidneys from seven different patients obtained during post-mortem examination and used for this study. All tissues underwent fixation in 10% buffered formalin. Tissue collection included samples from four kidneys from patients with confirmed germline mutations of the VHL gene and established clinical diagnosis of VHL disease (designated as “VHL kidneys”). All VHL patients had additional VHL disease-associated tumors including hemangioblastomas, epididymal cystadenomas, endolymphatic sac tumors, and microcystic adenomas of the pancreas. Three patients were male and one female. The causes of death were pneumonia in two cases, metastatic renal cell carcinoma, and intracerebral hemorrhage.

The VHL patients’ ages at the time of demise were between 50 and 65 years (average 58.5 years). Three control kidneys from patients with sporadic renal cell carcinoma, aged between 55 to 70 years (average 63.33 year), were also used for this study designated as “sporadic kidneys”.

All kidneys were sectioned randomly into maximally 10 cm × 1.5 cm × 1.0 cm columnar cuboids. The obtained cuboids of each kidney were inspected, and cuboids showing any grossly visible tumor were rejected. Of the remaining cuboids, the cuboid with the least grossly visible pathologic changes was identified and selected. For each kidney under study, one cuboid was selected (*n* = 7). Cuboids were then sliced into 0.3 cm × 1.5 cm × 1.0 cm blocks and a maximum of 30 blocks per cuboid was created. All tissue blocks without grossly visible pathologic change were paraffin embedded. An initial section was taken from each block and stained with hematoxylin and eosin (H&E) for histologic evaluation and identification for 3D reconstruction.

### 2.2. Immunohistochemistry

All immunohistochemical studies were stained using the Dako Flex-hrp system and developed with DAB. CAIX (Cell Marque 379R-18) underwent antigen retrieval in a high pH buffer followed by 20 min incubation with the primary mouse monoclonal anti-human CAIX antibody. CD34, a mouse monoclonal from Agilent/Dako IR632, underwent antigen retrieval in low pH, then the antibody was incubated for 15 min, the linker 10 min, and the Flex-hrp for 10 min. HIF is a rabbit polyclonal antibody from Novus Biological (NB100-122). The Antigen retrieval was low pH. The antibody was diluted 1:50 and incubated 30 min. All immunohistochemical studies were developed with DAB for 10 min and counterstained with hematoxylin. All H&E and immunohistochemical slides were digitized and analyzed using ImageScope, version 12.1.0.5029. Vewport 12.1.3. Copyright © Aperio Technologies, Inc. 2003–2013.

### 2.3. 3Dimensional Analysis

We recently described an algorithm for tissue procurement, processing, and 3D analysis of tissue blocks of interest [23]. The histopathologic examination was performed on normal-appearing renal parenchyma with minimal gross pathology and relatively well-preserved histologic details. We investigated 10 serially sectioned VHL kidney blocks from 4 different patients and three serially sectioned control blocks (normal kidney tissue of patients with sporadic carcinoma). All the H&E-stained slides were scanned and digitized at 50-micron intervals. We investigated and characterized the lesions which revealed clear cell amplification. Since the microscopic clear cell proliferations follow a predictable pattern, all pathologic clear cell events in this study have been divided into the following types:

1-Chain: multiple adherent clear cells that fill more than 50% of a tubular circumference.

2-Cluster: Collection of clear cells expanding and filling a renal tubule.

3-Complex cluster: Multiple profiles of intratubular aggregates of clear cells next to each other, with possible growth outside the tubules.

### 2.4. First Quantitative Approach Applied

The quantification was confined to chains, clusters, and complex clusters. First, we annotated all lesions with the letter C with subsequent numbers (x) in each investigated block (C1; C2; C3, and so on). Subsequently, we followed those lesions into the following H&E-stained slides which are 50 microns deeper than the previous slide. If the lesion (C1) still showed in that second slide, the lesion retained the same annotation of (C1), but we annotated that lesion as (C1-2), and if the other lesion (C2) still showed in first, second and third H&E-stained slide, it was annotated as (C2-3) and so on (where the number after the dash designates the number of H&E-stained slides in which the lesion was detectable).

### 2.5. Second Quantitative Approach Applied

Knowing that the distance between H&E-stained slides was 50 microns we could calculate the lesion’s height within the lumen of the renal tubule and divide them as follows:

Lesion 1 is equivalent to Cx-1 and is less than 50 microns in height in the tubular lumen.

Lesion 2 is equivalent to Cx-2 and is less than 100 microns in the tubular lumen.

Lesion 3 is equivalent to Cx-3 and is less than 150 microns in the tubular lumen.

Lesion 4 is equivalent to all lesions Cx-4 and exceeds 150 microns in height in the tubular lumen. Figure 1 illustrates the appearance of these lesions within the tissue block and how they would appear on different H&E-stained levels.

### 2.6. Third Quantitative Approach Applied

We applied the following equations only to cluster lesions, and it represents a measurement of the volumes of lesions in cubic mm. The volume was calculated using a conical frustum as a model. The studied lesions were presumed to have a frustum shape. We measured the lesions by drawing a circle around the center of the lesion in a way that it approximated the area of the lesion. This circle annotation surface area was used as an equivalent to the lesion’s surface area, and then we measured the radius of that circle. Subsequently, we measured the radius of the same lesion in the 50 microns deeper from the previous slide Figure 2A is an example of the radius measurement at three different H&E-stained levels that are 50 microns apart. Finally, we used the formula to calculate the Frustum volume [Volume of a conical frustum: V = (1/3) × π × h × [r1^2^ + r2^2^ + (r1 × r2)], and we did add to the calculated frustum volumes two pyramid volumes, one on the top of the frustum and another on the bottom, with the assumption that all our lesions extend above and down by 25 μm, using the formula of pyramid volume (V = (1/3) × π × h × r^2^). The lesion 2 volume (Cx-2) = one frustum volume + 2 pyramids volumes. Lesion 3 (Cx-3) volume= 2 frustum volumes + 2 pyramids volumes. Figure 2B is a diagram explaining the volume measurement technique applied in this study.

## 3. Results

The previous study [23] showed that most pathologic events in VHL patients were single clear cells, followed in lower frequency by chains, and the least frequent were complex or invasive clusters. In our control cases (normal kidney parenchyma of sporadic patients with resected renal cell carcinoma) only six to 12 isolated clear cells were identified. We did not detect any clear cell clusters in our three sporadic, serially sectioned control kidneys.

### 3.1. Lesions Account

After inspecting and annotating all the lesions, the total number of lesions visualized and annotated in the H&E-slides from 10 blocks of 4 VHL patients studied were 1194 chains, clusters, and complex clusters. The studied consecutive sections had a distance of 50 micrometer from each other. Those are subsequently subclassified as:

Lesion 1 (Cx-1): 958 chains or clusters (lesion present in one section only, not in any consecutive section).

Lesion 2 (Cx-2): 154 chains or clusters (lesion present in two consecutive sections only).

Lesion 3 (Cx-3): 53 chains or clusters (lesion present in three consecutive sections only).

Lesion 4 (Cx-4): 29 chains or clusters (lesion present in four consecutive sections only).

The total number of lesions studied in each block of all VHL patients is demonstrated in Table 1. The frequency pattern of lesions (the most frequent is lesion 1, then 2, 3, and the least are 4) is illustrated in Figure 3.

Table 2 summarizes the number of lesions and their percentages in the studied blocks.

### 3.2. Immunohistochemical Expression

Regarding the immunohistochemical studies, sixteen slides from seven blocks of four patients with VHL disease were stained with CAIX antibody. CAIX-positive cells were observed in all clear cell changes in all stained slides regardless of the size (small chains, big chains, clusters, and invasive clusters). Figure 4A shows representative images of the expression of CAIX in variably sized lesions. A CD34 stain was performed as an endothelial marker on sixteen slides from three blocks of two different VHL patients to detect the degree of vascularization. We observed a substantial increase in the extent of vascularization around tubules containing clear cells. The vessels were most evident around complex lesions, with a gradual decrease in the number of observed vessels in clusters followed Figure 4B. This marked increase in vascularization around clear cell lesions (CAIX-positive foci) shows that they promote angiogenesis.

### 3.3. Cluster Volumes

We calculated the volumes of all clusters examined, which are 119 in total. We found that they range in volume from 0.000034–0.0671 mm^3^. We subsequently subcategorized them into three groups from smallest volume to largest volume. In category one, there were 105 lesions (volume range 0.000034–0.00096 mm^3^). Category two, there were 10 lesions (volume range 0.00131–0.00956 mm^3^). Category three had four lesions (volume range 0.0102–0.0671 mm^3^).

## 4. Discussion

Our study demonstrates that careful microscopic examination of grossly normal kidney tissue in VHL patients revealed numerous clear cell chains and clusters in comparison to grossly normal kidney tissue in patients with sporadic clear cell renal cell carcinoma. We investigated the multicellular foci in the grossly normal cortex of VHL kidneys and counted them in 10 blocks of 4 different VHL cases. We identified that the count of these pathological foci was sequentially decreased from lesion 1 to lesion 2, then from lesion 2 to 3, and finally, the lowest was lesion 4, in each case, and each block studied. Distribution was highly consistent in the four cases. The same sequential decrement was noticed in clusters volumes, too: the most frequent category 1 (105 clusters), category 2 (10 clusters), and the least was category 3 (4 clusters).

Given that we have cut through (i.e., exhausted the tissue) the tested blocks, H&E stained them, and performed a 3dimensional analysis of these structures by which approach we establish certainty that clear cell proliferations do not represent extensions of larger pathologic processes that were invisible on the original section. We primarily analyzed all H&E-stained sections for structures of interest, followed by examining expression of CAIX and CD34.

All samples from different multicellular clear cell foci examined were positive for CAIX, and it is well-established to detect activation of the HIF pathway. Expression of the *CAIX* gene is regulated by the VHL/HIF system via a HIF responsive element in the promoter, and CAIX constitutes an attractive marker for HIF activation because of the inducible response and the stability of the protein [24,25].

Mandriota et al. [9] confirmed that the foci of CAIX expression represented activation of the HIF pathway. They examined the expression of two other HIF-responsive genes encoding GLUT-1 [26] and VEGF [27]. Immunolabeling for GLUT-1 and in situ-hybridization for VEGF revealed high levels in overt CCRCC and collections of cells in the non-tumorous renal parenchyma that corresponded with the CAIX-expressing foci. As a marker for angiogenesis, CD34 expression was analyzed to observe a substantial increase in the extent of vascularization around tubules containing CAIX-positive cells.

One observation we had in this study after performing CD34 immunohistochemical stain is the proportional increase in the density of vasculature with the increment of the size of clear cell lesions from chains to clusters, to invasive foci. This observation supports the fact that these clear cell foci promote angiogenesis, which allows tumor growth and possible angiogenic invasion with continuous growth.

Given the findings in our studies, and since the microscopic clear cell proliferations follow a predictable pattern, we hypothesize that the renal neoplastic pathological events (i.e., atypical cysts and renal cancers) evolve from a variety of microscopic precursors (i.e., clear cells aggregates and clusters).

Mubarak et al. showed that the most frequently observed pathologic events in VHL kidney sections were either isolated clear cells within a renal tubule or two or more clear cells adherent to each other, likely representing changes secondary to an amplification event. More significant accumulations of clear cells along the tubular lining, as well as clustering of clear cells within renal tubules, appeared to be in the continuity of smaller preceding events [23].

Clear cell changes in renal tubules are not always neoplastic and can be seen secondary to metabolic or toxic etiologies (e.g., toxins, Alport nephropathy, osmotic tubulopathy secondary to radiographic contrast) [28,29,30]. In contrast to the clear cell changes seen in VHL patients, those changes are seen in cells displaying a brush border or tubular form of differentiation. Additionally, immunohistochemically the clear cells in VHL cases are consistently and reliably immunoreactive for anti-CAIX antibody [23,31].

In other studies in patients with VHL disease, it has been shown that careful microscopic examination of grossly normal tissue of the spinal cord and epididymis revealed precursor structures that are numerically more frequent than tumors [15,32]. Spinal cord tissue from patients with VHL disease presumed to be ‘tumor-free’ harbors microscopic foci of poorly differentiated cellular aggregates in nerve root tissue. Additionally, they showed that a small subset of VHL-deficient microscopic lesions extends beyond the nerve root to form early hemangioblastoma. Intraradicular precursors consist of scattered VHL-deficient cells with activation of HIF-2 alpha and HIF-dependent target proteins, including CAIX and VEGF and are associated with an extensive angiogenic response. Ultrastructural examination revealed the gradual transition from poorly differentiated VHL-deficient cells into vacuolated cells with a ‘stromal’ cell phenotype. The evolution of hemangioblastoma seems to involve multiple steps from a large pool of precursor lesions [33].

Shively et al. showed that in surgically resected nervous system hemangioblastomas of patients with VHL disease, HIF-1 alpha activation was associated with epithelioid structures, which were mainly seen in larger tumors. However, small tumors were composed mostly of poorly differentiated mesenchymal structures and did not show HIF-1 alpha activation. This suggests that the growth of nervous system tumors in VHL disease has a sequence of structural and molecular events [32].

The fact that the microscopic renal cystic and solid neoplasms containing only clear cell cytological features were found in patients with VHL disease [13] gave us a clue that those microscopic foci might play a role in the development of those neoplastic pathological lesions and a role in tumorigenesis.

Of particular interest in VHL disease has been the relationship between cysts and tumors. Additional studies may better clarify whether cysts and tumors develop via independent mechanisms or whether they are merely variations of an essentially uniform pathogenetic process. While consequences of dysregulation of VHL and HIF pathways provide numerous potential clues [34], our approach may be helpful in obtaining in-situ evidence in human tissues in the future.

Our morphological data showed different types of lesions which represent intriguing candidates for progression into cystic lesions and tumors in the kidney. The genetic and epigenetic mechanisms for these progressive events remain to be clarified. Walther et al. investigated normal-appearing renal tissues from VHL patients [22]. They detected an abundance of small clear cell tumorlets in VHL kidney; single H&E-stained sections from a multitude of paraffin blocks had been investigated to document and extrapolate the frequency of tumorlets, which was estimated as 600 per VHL kidney.

While previous studies by Mubarak et al. had been concerned with lesion quantification only, this study focuses on the larger simple and complex clear cell clusters, with more detailed information on their volumetric extent. Here we were able to not only show inverse correlation between lesion complexity and frequency, but—through 3D approach—between lesion size and frequency which we believe to represent the histomorphologic equivalent of linear “evolutionary tumorigenesis”, recently postulated for VHL disease-associated clear cell carcinoma after whole genome analysis of a series of renal carcinomas [35].

Our findings demonstrate more than 1000 multicellular amplification events in less than one cubic cm of “normal” kidney tissue, the largest one measuring less than one thousandth of a cubic mm, undetectable by naked eye or radiologically. In the patients under study, VHL disease in human kidney therefore presents as wide-spread substitution of tubular cells with clear cells, only a small subset of which will undergo amplification into the next level of complexity. As we show here, this principle of protraction directly applies to lesion size.

Previously, this principle of protraction has been radiologically documented in far more than 1000 times larger, radiographically detectable lesions [36]. In conjunction with the “NIH rule” that tumors smaller than 3cm almost never metastasize (reviewed in [36]), this led to current therapeutic principles of pure observation of smaller nodules and nephron-sparing surgery at the time of surgical intervention.

Our findings may encourage diagnostic and academic pathologists to not only pay attention to primary tumor pathology in resection specimens, but also to adjacent normal kidney tissue; the quantity of our described precursor structures may give information on the “activity” of VHL pathology in these tissues, and may have prognostic implications for the likelihood of new tumor formation. Of particular current therapeutic interest is molecular targeting through immunotherapy [12], and resection specimens that are subjected to our approach may give new insight on immunotherapeutic efficacy on precursor structures. While the required serial sectioning of normal kidney tissue blocks is cumbersome, this activity can be automated by serial sectioning robots that have been recently developed [37]. Furthermore, advances in artificial intelligence development would allow to automatically scan for these relatively uniform precursor structures allowing for generation of quantitative data within reasonable time frames. Artificial intelligence analysis of 3D imaging data is rapidly developing; it has, e.g., shown encouraging results in limited-resection specimens of lung cancers [38], and we do not see any obstacles applying such technologies to analysis of “normal” tissues.

Finally, microdissection of precursor structures combined with targeted genetic analysis may give insight into earliest genetic changes leading to progression of precursors from single cell to complex cluster; 3D reconstruction could also be applied to serially sectioned frozen tissue blocks of normal VHL kidney allowing for mass spectroscopic analysis for earliest proteomic change after microdissection, and applicability of proteomics for diagnosis of primary VHL tumors has been previously demonstrated [39].

The main limitation of our study is the small number of cases tested. However, this study has examined sequential levels cut at 50 microns (from a total from 10 blocks of 4 VHL patients). Each slide was digitized and annotated accordingly. Each annotation was followed in each cross -section to accurately assess the frequency, sizes, and volumes of these clear cell chains and clusters occurring in the grossly normal kidneys of patients with VHL. Similar studies in the literature are lacking.

## 5. Conclusions

The study of morphological changes and histogenesis in VHL kidneys leads to a better understanding of the sequence of tumorigenesis. Additionally, this study has shown us the prevalence of these clear cell changes and sublocalization within the kidney parenchyma to be in close proximity to a certain anatomic/histologic structure in the kidney, which is leading us to a conclusion regarding the cell of origin in clear cell renal cell carcinoma [40]. Further studies are needed on the role these different microscopic and molecular events play in renal carcinogenesis.

## Figures and Tables

**Figure 1 medicina-58-01725-f001:**
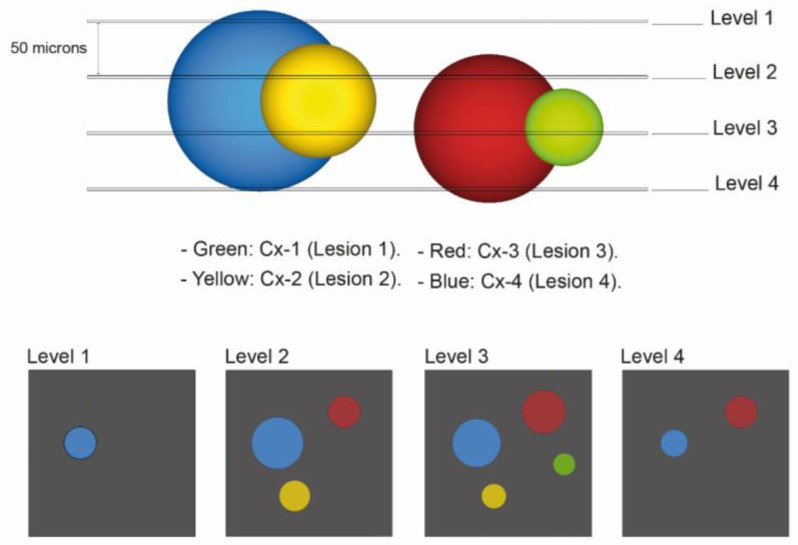
Diagram of VHL kidney block containing pathologic events of different sizes which are displayed in different colors. For the first quantitative approach, lesion annotation was performed, with retaining the annotation of a lesion that reoccurs in the next deeper level. The histologic presentation of different H&E-stained section levels into the block is shown (annotated as “Level 1”, “Level 2”, “Level 3” and “Level 4”). The second quantitative approach was concerned with identification of the exact number of levels lesions were present in: Green: Cx-1 (lesion 1): all chains or clusters which were seen on one H&E-stained slide only. Yellow: Cx-2 (lesion 2): all chains or clusters which were seen on two serially H&E-stained slides. Red: Cx-3 (lesion 3): all chains or clusters which were seen on three serially H&E- stained sections. Blue: Cx-4 (lesion 4): all chains or clusters which were seen on four serially H&E- stained sections and above.

**Figure 2 medicina-58-01725-f002:**
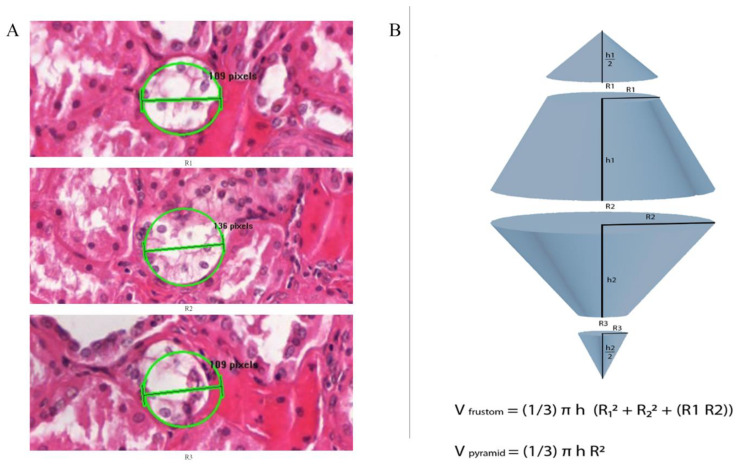
(**A**) Representation of H&E-stained slide illustrating diameter (2r) measurement for cluster lesion. First, we drew a circle over the lesion of approximately same area. Then we measured the radius of that circle which represents the lesion’s radius, then measured the radius of this lesion in the tenth serial section, which is 50 microns from the previous slide. (**B**) Diagram illustrates the formula to measure the Frustum volume, V = [(1/3) × π × h × (r1^2^ + r2^2^ + (r1 × r2)]; we added the volume of two pyramids, one on the top and the other on the bottom of the cluster by using the formula of pyramid volume (V = (1/3) × π × h × r^2^). Volumes of lesion 2 (Cx-2) were measured by measuring the volume of one frustum R1 and R2, plus two pyramid volumes, where R1 and R2 are the radius of above and bottom pyramids respectively of 25 microns height. Volumes of lesion3 (Cx-3) were measured by calculating the volume of two frustums, plus two pyramid volumes, where R1 and R3 are the radius of above and bottom pyramids respectively of 25 microns height.

**Figure 3 medicina-58-01725-f003:**
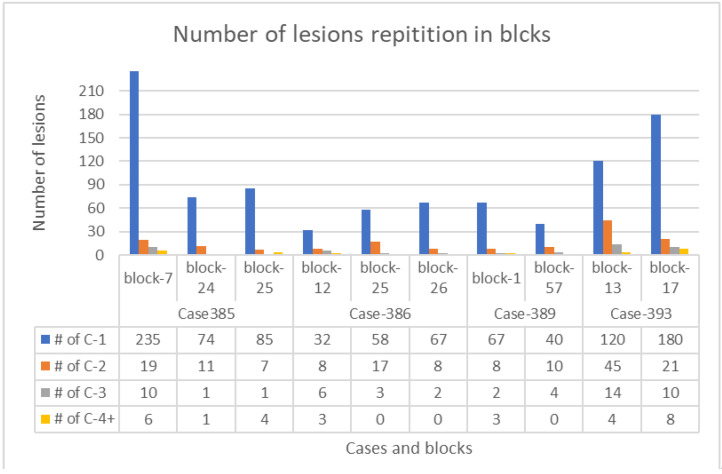
A lesion column chart showing the number of all types of lesions in each block from the four VHL patients. Blue columns—lesion 1 (Cx-1), orange columns—lesion 2 (Cx-2), gray columns—lesion 3 (Cx-3), and yellow columns—lesion 4 (Cx-4).

**Figure 4 medicina-58-01725-f004:**
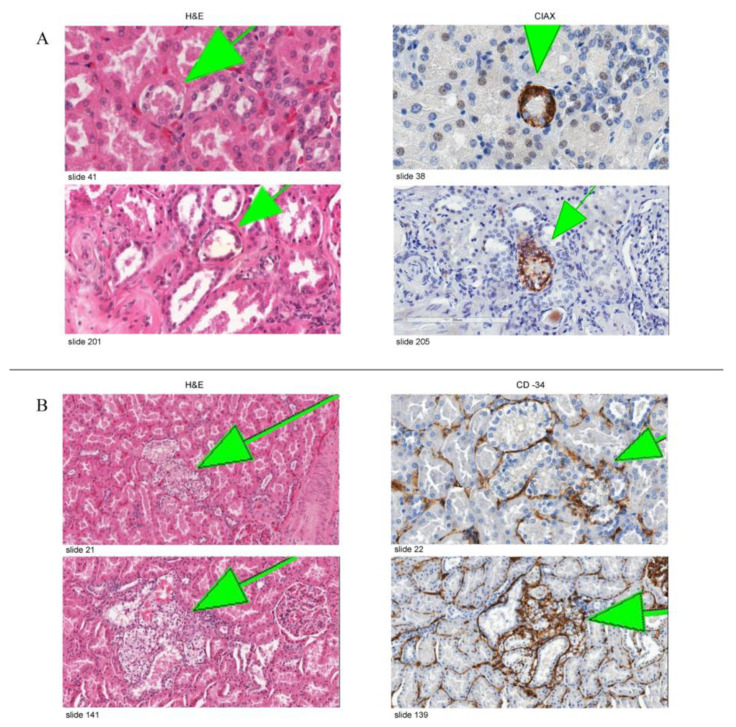
(**A**) Representative images of CAIX staining in VHL kidneys. The left column shows two lesions on H&E-stained slides, and the right column shows a serial section of the same lesion stained with anti-CAIX antibody. (**B**) Representative images of CD34 staining in VHL kidneys. Figure demonstrating a gradual increase in the density of the vasculature around the clear cell changes with the increment of the volume.

**Table 1 medicina-58-01725-t001:** Number of lesion repetition in blocks. The number in lower column represents the total number of all lesions annotated and analyzed (chains, clusters, and invasive clusters) in 10 blocks from 4 cases of VHL patients.

Case 385			Case 393		Case 389		Case 386		
Block #7	Block #24	Block #25	Block #13	Block #17	Block #1	Block #57	Block #12	Block #25	Block #26
270	87	97	183	219	80	54	49	78	77

**Table 2 medicina-58-01725-t002:** Prevalence of multicellular microscopic lesions. Prevalence of studied multicellular microscopic lesions in the normal cortex of kidney of patients with VHL disease.

Type of Lesions	Lesion 1	Lesion 2	Lesion 3	Lesion 4	Total
**Number of lesions in all slides**	958	154	53	29	1194
**Percentage of each type from total**	80.2%	12.9%	4.4%	2.5%	100%

## Data Availability

All data supporting this study’s findings are available with correspondents upon request.

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
