# Peer review of "Characterization of Microscopic Multicellular Foci in Grossly Normal Renal Parenchyma of Von Hippel-Lindau Kidney"

_medicina, 2022, doi:10.3390/medicina58121725_

Round 1
Reviewer 1 Report (Previous Reviewer 1)
To the authors
Thank you for the opportunity to review the revised manuscript. The comments raised by the reviewer are well addressed.
Author Response
Thank you for reconsidering our manuscript (2023015) for publication.
Reviewer 2 Report (Previous Reviewer 2)
This study aims to describe the earliest renal lesions in patients with von Hippel-Lindau (VHL) disease, especially the multicellular microscopic pathologic 21events, to get information into the genesis of renal neoplasms in this condition.
Nevertheless, despite the overmentioned strengths, there are some limitations that affect the quality and accuracy of the paper.
Therefore, I would suggest reviewing this manuscript for publication.
Specifically,
Personally I think that all the aspects the authors explored should have a clinical impact in order to improve both diagnostic annd therapeutic pathways. For this reason, I would suggest the authors to add a paragraph in the section discussion where they can discuss about clinican implications and possibile applications in the era of new technoligies.
For the same reason, It would be very interesting to insert in the manuscript as future perspective how this findings might mix with new technologies improving diagnosis and prognosis of this rare condition ( the authors could add these references that explain how technoligies applied to clinical conditions play a pivotal role in different clinical scenarios
10.23736/S0393-2249.20.03706-6; 10.23736/S0393-2249.18.03278-2)
Author Response
Please see the attachment.

This manuscript is a resubmission of an earlier submission. The following is a list of the peer review reports and author responses from that submission.
Round 1
Reviewer 1 Report
To the authors
In this study, the authors investigated the sequence of events from the earliest pathologic change in VHL kidney to the neoplastic stage and also showed the usefulness of CAIX and CD34 expression. This report is very interesting. However, several problems should be resolved to be accepted for publication in the Medicina.
1. In figure 4, to confirm the existence of vascularization around CAIX-positive cells, the authors should show immunohistochemistry using serial section methods or n immunofluorescent study.
2. There is limited data about the patient’s background, and the authors should add information about what kinds of VHL disease-associated tumors patients had. Furthermore, data relevant to RCC, such as hematuria and tumor markers, are also needed.
Overall this is the well-written manuscript.
Reviewer 2 Report
The authors should be congratulated for the work, however I have several concerns:
1. What do you think they are the additional information provided by this study that has not been already clarified by your previous one? Apart from the slight focus on the immunohistochemical analysis showing the well known existing relationship between VHL and CAIX expression, what you observed and described in this paper has already been exstensively illustrated by Mubarak et al study. Therefore you should mention these limitation and underline what are instead the point of strenght of this study.
For example you can stress the importance of having a better understanding of pathogenesis, natural history and follow-up of these kind of patients expecially because of the delicate clinical and surgical implication: infact these patient are at hight risk of developing other several neoplasia [10.1007/s00345-020-03574-5], and at high risk of recurrent renal cell carcinoma which means undergoing several surgical interventions that could potentially lead to CKD. In this scenario, stressing the importance of a nephron sparing approach is crucial [10.1007/s00345-019-02879-4].
2. The first quantitative approach applied is not well explained and maybe need a visual clarification. At the contrary, the third quantitative approach is very intersting and innovative, but what does it add to the main goal of your paper? Is there an association between the cluster volume and CAIX/CD34 expression? You could also focus on explaining how the 3d model is taking place and how it can help in several occasion, expecially when a visual layout is needed. [10.3389/fsurg.2021.665328]
3. Even though it is described in your previous study, I suggest you to explain in materials and methods the sampling techinique you used when processing these kidneys, otherwise there is a step missing that makes impossibile to understand this paper.
4. The abstract is not clear and its conclusion doesn’t match with the actual aim of the study.